# Peptide vaccine-treated, long-term surviving cancer patients harbor self-renewing tumor-specific CD8+ T cells

Eishiro Mizukoshi [1,2✉], Hidetoshi Nakagawa[1,2], Toshikatsu Tamai[1], Masaaki Kitahara[1], Kazumi Fushimi[1], Kouki Nio[1], Takeshi Terashima[1], Noriho Iida [1], Kuniaki Arai[1], Tatsuya Yamashita [1], Taro Yamashita[1], Yoshio Sakai[1], Masao Honda [1] & Shuichi Kaneko [1]

The behaviors and fates of immune cells in cancer patients, such as dysfunction and stem-like states leading to memory formation in T cells, are in intense focus of investigation. Here we show, by post hoc analysis of peripheral blood lymphocytes of hepatocellular carcinoma patients previously undergoing vaccination with tumour-associated antigen-derived peptides in our clinical trials (registration numbers UMIN000003511, UMIN000004540, UMIN000005677, UMIN000003514 and UMIN000005678), that induced peptide-specific T cell responses may persist beyond 10 years following vaccination. Tracking TCR clonotypes at the single cell level reveals in two patients that peptide-specific long-lasting CD8+ T cells acquire an effector memory phenotype that associates with cell cycle-related genes (*CCNA2* and *CDK1*), and are characterized by high expression of *IL7R*, *SELL*, and *NOSIP* along with a later stage promotion of the AP-1 transcription factor network (5 years or more past vaccination). We conclude that effective anti-tumor immunity is governed by potentially proliferative memory T cells, specific to cancer antigens.

---

[1] Department of Gastroenterology, Graduate School of Medicine, Kanazawa University, 13-1 Takaramachi, Kanazawa Ishikawa 920-8641, Japan. [2]These authors contributed equally: Eishiro Mizukoshi, Hidetoshi Nakagawa. ✉email: eishirom@m-kanazawa.jp

Over the past two decades, the roles and mechanisms of the immune system in controlling cancer have been clarified[1,2] and used for novel therapeutics that have resulted in enormous success in treating cancers. Cancer immunotherapy approaches are classified into several classes. For example, immunomodulators including immune checkpoint inhibitors to enforce effector immune functions[3–5], vaccines that educate the immune system to elicit anti-cancer activity[6], and cell-based immune therapies of recent sophisticated cell transfers, such as chimeric antigen receptor (CAR)-T cells[7], are current topics of cancer therapeutics. Because the main effectors that mediate enhanced antitumor immune responses by these therapies are CD8[+] cytotoxic T lymphocytes (CTL), it is important to pursue the leading factors that affect CTL functions to further elucidate and develop current immunotherapies and formulate novel immunotherapeutics against cancer.

Peptide vaccines against cancer can augment tumor-specific T cell responses. Numerous peptides induce tumor-specific CD8[+] CTLs and tumor-specific helper CD4[+] T cells in association with class I and II major histocompatibility complex (MHC) molecules, respectively[8,9]. These peptides have been identified among tumor-associated antigens (TAA), cancer-testis antigens, and neoantigens. We have tested various TAA-derived peptides for treatment of hepatocellular carcinoma (HCC) in a series of phase I clinical trials[10–13] and observed effective CTL responses and durable clinical benefits with corresponding functional T cells and T cell receptors (TCR) in some patients that have received the peptides[14]. However, in most patients, the magnitudes of CTL responses were low, which might hinder clinical outcomes. Unsatisfactory CTL induction by peptide vaccines can be improved. Administration of multiple peptides that cover more than two antigens can induce effective CTL responses and may compensate for tumor immune escape and heterogeneity[15]. Addition of CD4 epitope peptides has also been investigated to enhance CD8[+] CTL responses, although the conclusion remains controversial[16,17]. Most recently, high-throughput genomic analysis and progress in epitope prediction have enabled the design of personalized epitope peptides on the basis of mutations in cancer, which may be able to overcome diverse phenotypes of cancer between individuals[18].

In addition to CTL induction efficacy, another important aspect is obtaining durable immune responses. However, to date, most clinical trials of cancer vaccines have rarely followed up the immune responses beyond 1 year.

In this study, we report persistent immune responses for 10 years after vaccination by tracking the TCR clonotypes with their phenotypes at the single-cell level and define the factors that are required for sustainable functions of tumor-specific T cells.

## Results

**Peptide-specific CTL induction after peptide vaccination.** We conducted a post hoc observational study of the five phase I clinical trials that had tested peptide vaccines for hepatocellular carcinoma, which included 14, 12, 12, 15, and 12 patients who were treated with hTERT$_{461}$ (UMIN000003511, recruitment: December 2009–September 2010), SART2$_{899}$ (UMIN000004540, recruitment: December 2010–March 2011), SART3$_{109}$ (UMIN000005677, recruitment: June 2011–March 2013), AFP$_{357}$/AFP$_{403}$ (UMIN000003514, recruitment: June 2010–March 2012), or MRP3$_{765}$ (UMIN000005678, recruitment: June 2012–August 2014) peptides, respectively (Supplementary Table 1). The former three trials were in combination with local ablation therapy for early-stage HCC and the latter two trials were for intermediate or advanced stages of HCC as a monotherapy or following hepatic arterial infusion chemotherapy, respectively[19,20]. These trials were

carried out at Kanazawa University. The patient demographics and clinical characteristics have been reported previously[10–13] (Supplementary Table 2). The HCC stages and serum AFP levels in the AFP and MRP3 trials were more advanced than those in the other trials. Other clinical factors, including age, sex, etiology of liver disease, liver histology, liver functions, and serum ALT levels, were not significantly different among the trials. We first examined cytotoxic T cell induction as judged by the IFN-γ ELISpot assay by comparing PBMC samples collected before (pre) and at 3–6 months after (post) vaccination (Supplementary Table 3). In the A2 case from the hTERT trial, immunospots were rarely seen in both pre and post samples without a peptide, whereas for the hTERT$_{461}$ peptide, the post sample displayed numerous immunospots that did not occur on the membrane of the presample (Fig. 1a). In D1 samples of the AFP trial, AFP$_{357}$- and AFP$_{403}$-specific immunospots had also developed after vaccination (Fig. 1b). We defined positive cytotoxic T cell (CTL) induction [CTL(+)] as a ≥10 specific spot increase and two-fold increase to identify CTL(+) cases in each trial. CTL(+) cases in hTERT$_{461}$, SART2$_{899}$, SART3$_{109}$, AFP$_{357}$/AFP$_{403}$, and MRP3$_{765}$ trials were observed in 8 out of 14 (57.1%), 4 out of 12 (33.3%), 4 out of 12 (33.3%), 5 out of 15 (33.3%), and 4 out of 12 (33.3%), respectively (Fig. 1c, d).

**Long-term survivors marked by peptide-specific immune induction.** We conducted the post hoc long-term follow-up study of the participants in the five peptide vaccine trials. We show tables of individual survival data with their peptide-specific CTL responses (Tables 1–5). To address whether the vaccination-induced irrelevant immune responses in addition to the peptide-specific immune reactions, we also examined Cytomegalovirus (CMV)-derived pp65$_{328}$-specific immune induction using the same criteria as used in the peptide-specific CTL induction (Supplementary Fig. 1). Contingency analyses of installations of the CMV-specific and the vaccine-specific responses resulted in no relation between these two strains of immune inductions in any of the trials (Tables 1–5). Next, we focused on the individuals who survived long periods. Remarkably, patient D3 who had suffered from sorafenib-refractory advanced HCC finally survived for more than ten years after the AFP vaccine administration. In the hTERT vaccine trial, A2, A4, A6, and A8 stayed alive for 10 years after the adjuvant hTERT immunization following the curative ablation. In the other trials, B7 and C4 also achieved durable survivals; 8.7 years and 9.2 years, respectively. We note that the shared feature of these patients was positive CTL induction, suggesting its involvement in the long-term maintenance of antitumor immunity. To understand the mechanisms that brought durable survival periods in these participants, we were asked to further interpret and trace the induced immune responses over the observational periods.

**Proliferation of peptide-specific CD8[+] T cells with an effector memory phenotype by the vaccinations.** In addition to ELISpot assays, we also evaluated peptide-specific T cell induction by flow cytometry using hTERT$_{461}$, AFP$_{357}$, AFP$_{403}$, and MRP3$_{765}$ tetramers along with other phenotypic markers (see Supplementary Fig. 2). SART2$_{899}$ and SART3$_{109}$ tetramers were unavailable because of refolding issues during monomer synthesis. In the A2 case, the hTERT$_{461}$ tetramer-binding CD8[+] T cell population expanded from 0.03% to 0.14% of CD8[+] T cells after vaccination (Fig. 2a). Not only the percentages of tetramer[+] cells, but also the phenotypes were different. In the pre sample, the CCR7[−]CD45RA[+] terminally differentiated effector memory phenotype (TEMRA) was the dominant phenotype (63.3%) among the tetramer[+] cells. However, the CCR7[−]CD45RA[−] effector memory (EM) fraction

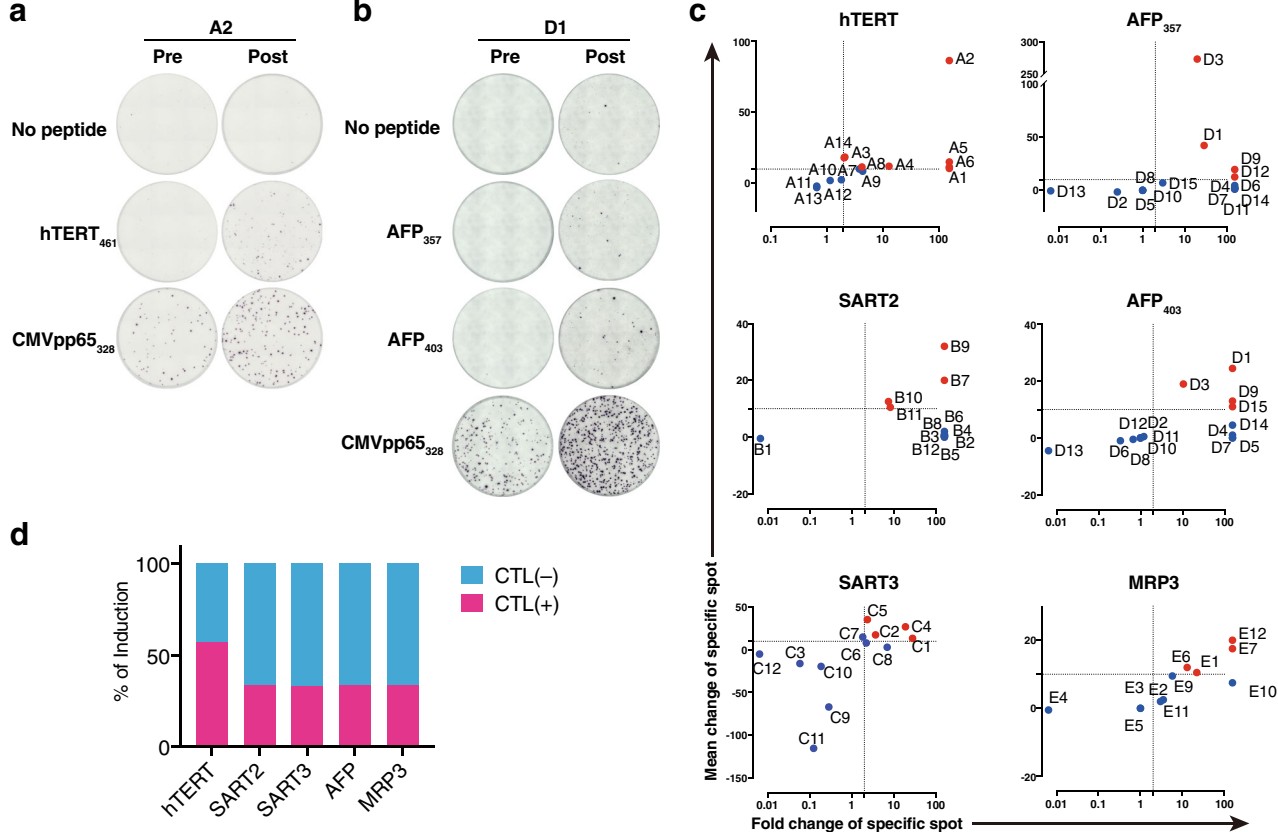

**Fig. 1 Peptide-specific immune responses induced by peptide vaccine treatments.** PBMCs were collected before and at 3–6 months after vaccination (pre and post, respectively) from each participant and tested to assess antigen-specific IFN-γ production by an ELISpot assay. Patients from the five clinical trials were tested. Representative images of the IFN-γ ELISpot assay of an hTERT-peptide vaccine patient (case A2) and an AFP-peptide vaccine patient (case D1) are shown. In the A2 experiment, PBMCs collected at two time points were examined and IFN-γ secretion in response to either no peptide, $hTERT_{461}$ peptide (test peptide) or CMV (cytomegalovirus) $pp65_{328}$ peptide (positive control) were displayed in the round membranes. Each colored spot was considered to be a cell that produced IFN-γ (**a**). In the D2 experiment, IFN-γ production against no peptide, $AFP_{357}$, $AFP_{403}$, or $CMVpp65_{328}$ peptide is presented (**b**). Immune responses measured by the IFN-γ ELISpot assay are plotted in X-Y graphs with annotations of the patient IDs. The X-axis represents the fold change of specific spots and the Y-axis indicates an increase of the specific spot number by comparing pre and post samples. Because positive induction of peptide-specific cytotoxic T lymphocytes (CTLs) was defined as both a more than 10 increase and more than two-fold increase of the specific spot number, dotted lines are drawn at X = 2 and Y = 10 to distinguish cases with positive CTL induction. Positive CTL inductions are highlighted in red and negatives are highlighted in blue. If both pre and post were 0, X was defined as 1. If only pre was 0, the X-value was offset out of X = 100 and if only post was 0, the X-value was placed next to X = 0.1 (**c**). The proportions of positive CTL induction in each peptide study are shown in bar graphs. Positive CTL induction (CTL + ) is filled in magenta and negative CTL induction (CTL–) is shown in cyan (**d**). Source data are provided as a Source Data file.

became the majority in the post sample (64.1%), while no difference was seen in total CD8[+] T cell fractions (Fig. 2b). In $AFP_{357}$ and $MRP3_{765}$ tetramer staining, consistently enhanced tetramer frequencies were seen when CTLs were induced, whereas the $hTERT_{461}$ tetramer[+] percentages were increased overall after vaccination regardless of whether CTLs were induced. $AFP_{403}$ staining did not show any tendency (Fig. 2c). Combined analysis showed a significant increase of the tetramer-positive population in both CTL( + ) and CTL( − ) groups (Fig. 2d). Phenotypic analysis revealed that EM frequencies were expanded and TEMRA fractions had shrunk only when CTLs were induced (Fig. 2e). These results suggested that tetramer[+] T cell induction did not always indicate functional CTL induction and that EM formation among tetramer[+] T cells reflected functional antigen-specific T cells. However, in $hTERT_{461}$ vaccination cases, EM formation was seen in both CTL( + )and CTL( − ) groups (Fig. 2f). Therefore, we next investigated other factors that determined the functionality of $hTERT_{461}$-specific T cells, which contributed to favorable outcomes. We found higher expression of inhibitory receptors in $hTERT_{461}^{+}$ CD8[+] T cells compared with $hTERT_{461}^{-}$ CD8[+] T cells

at 2–4 weeks after vaccination. For example, in the A7 case, PD-1 (65.4%) and CTLA-4 (32.1%) expression by $hTERT_{461}$ tetramer[+] CD8[+] T cells was higher than that by tetramer[−] fractions (Fig. 2g). This analysis was performed in six patients of which PBMCs were available between 2 and 4 weeks after the completion of three doses of the vaccine. Here, we observed a clear negative correlation between PD-1 or CTLA-4 expression by $hTERT_{461}$ tetramer[+] CD8[+] T cells and prognosis (Fig. 2h, i).

**CTL responses are preserved for 10 years after vaccination.** IFN-γ ELISpot assays were conducted at 1, 5, and 10 years for CTL( + ) patients who were alive and able to provide PBMCs at each timepoint. For patients treated with hTERT, SART2, SART3, AFP, or MRP3-derived peptide vaccines, PBMCs were available at 1 year after vaccination in six, four, three, three, and one cases, respectively. Analyses at 5 years after the start of vaccination were possible in four, one, one, two, and zero cases. In this experiment, we performed the ex vivo ELISpot assay, in which PBMCs were tested directly, and the ELISpot assay using PBMCs pretreated with the corresponding peptides for 7 days. In the 1-year analysis,

**Table 1 Individual survival data of the hTERT peptide vaccine trial.**

| Patient | Observation Period (years) | CTL response | CMV response | Outcome |
|---|---|---|---|---|
| A1 | 5.2 | + | – | dead |
| A2 | 10.9 | + | + | alive |
| A3 | 6.1 | + | – | dead |
| A4 | 10.4 | + | – | alive |
| A5 | 7.3 | + | – | dead |
| A6 | 10.5 | + | – | alive |
| A7 | 3.6 | – | – | dead |
| A8 | 9.8 | + | – | alive |
| A9 | 4.5 | – | – | dead |
| A10 | 3.0 | – | – | dead |
| A11 | 0.8 | – | – | dead |
| A12 | 2.9 | – | – | dead |
| A13 | 3.0 | – | – | dead |
| A14 | 7.8 | + | – | dead |
| | | Fisher's exact P value >0.9999 | | |

**Table 4 Individual survival data of the AFP peptide vaccine trial.**

| Patient | Observation Period (years) | CTL response | CMV response | Outcome |
|---|---|---|---|---|
| D1 | 3.5 | + | – | dead |
| D2 | 0.2 | – | – | dead |
| D3 | 10.2 | + | – | alive |
| D4 | 0.2 | – | – | dead |
| D5 | 0.5 | – | + | dead |
| D6 | 0.5 | – | – | dead |
| D7 | 0.6 | – | – | dead |
| D8 | 0.5 | – | – | dead |
| D9 | 0.5 | + | – | dead |
| D10 | 1.0 | – | – | dead |
| D11 | 0.5 | – | – | dead |
| D12 | 0.8 | + | – | dead |
| D13 | 1.6 | – | – | dead |
| D14 | 0.3 | – | – | dead |
| D15 | 1.4 | + | + | dead |
| | | Fisher's exact P value >0.9999 | | |

**Table 2 Individual survival data of the SART2 peptide vaccine trial.**

| Patient | Observation Period (years) | CTL response | CMV response | Outcome |
|---|---|---|---|---|
| B1 | 5.5 | – | – | dead |
| B2 | 4.4 | – | + | alive |
| B3 | 7.3 | – | – | dead |
| B4 | 5.9 | – | – | dead |
| B5 | 7.8 | – | – | dead |
| B6 | 7.7 | – | – | dead |
| B7 | 8.7 | + | – | alive |
| B8 | 5.3 | – | – | dead |
| B9 | 2.7 | + | – | dead |
| B10 | 4.1 | + | – | dead |
| B11 | 6.1 | + | – | alive |
| B12 | 2.8 | – | – | dead |
| | | Fisher's exact P value >0.9999 | | |

**Table 5 Individual survival data of the MRP3 peptide vaccine trial.**

| Patient | Observation Period (years) | CTL response | CMV response | Outcome |
|---|---|---|---|---|
| E1 | 0.7 | + | + | dead |
| E2 | 1.1 | – | + | dead |
| E3 | 1.4 | – | – | dead |
| E4 | 2.6 | – | – | dead |
| E5 | 1.5 | – | + | dead |
| E6 | 1.5 | + | – | dead |
| E7 | 1.2 | + | – | dead |
| E8 | 0.1 | – | ND | dead |
| E9 | 0.3 | – | – | dead |
| E10 | 0.9 | – | – | dead |
| E11 | 1.6 | – | + | dead |
| E12 | 0.9 | + | – | dead |
| | | Fisher's exact P value >0.9999 | | |

**Table 3 Individual survival data of the SART3 peptide vaccine trial.**

| Patient | Observation Period (years) | CTL response | CMV response | Outcome |
|---|---|---|---|---|
| C1 | 3.7 | + | – | dead |
| C2 | 6.5 | + | + | dead |
| C3 | 7.7 | – | – | alive |
| C4 | 9.2 | + | + | alive |
| C5 | 1.2 | + | – | dead |
| C6 | 6.0 | – | – | dead |
| C7 | 8.2 | – | – | alive |
| C8 | 4.2 | – | – | dead |
| C9 | 4.3 | – | – | dead |
| C10 | 4.2 | – | – | dead |
| C11 | 7.2 | – | – | alive |
| C12 | 8.0 | – | – | alive |
| | | Fisher's exact P value 0.0909 | | |

positive ex vivo ELISpot results (> 10 spots) were only seen in A4 and D3, and not in SART2, SART3, or MRP3 analyses, while the pretreated PBMCs showed potent responses in all available participants of hTERT and the AFP trials, one patient (B11) in the SART2 trial, and two patients (C1 and C4) in the SART3 trial. No positive ELISpot result using pretreated PBMCs was obtained in the MRP3 study. In the 5-year analysis, only three samples were positive in the ex vivo ELISpot, which were all in the hTERT study (A2–A4). Pretreatment revealed stronger results from these three samples of the hTERT study and two more positive cases in the AFP study (D3) and SART3 study (C4) (Fig. 3a–e). The five patients who survived for 10 years were examined further, which resulted in detection of peptide-specific CTLs in all of them by ex vivo examination and in five of them after stimulation with the peptides for 7 days. These results suggested perfected memory formation of the peptide-specific T cells (Fig. 3f).

**Functional peptide-specific TCRs are preserved.** To dissect the immune responses that might contribute to a preferable prognosis at the peptide-specific T cell level, we analyzed individual

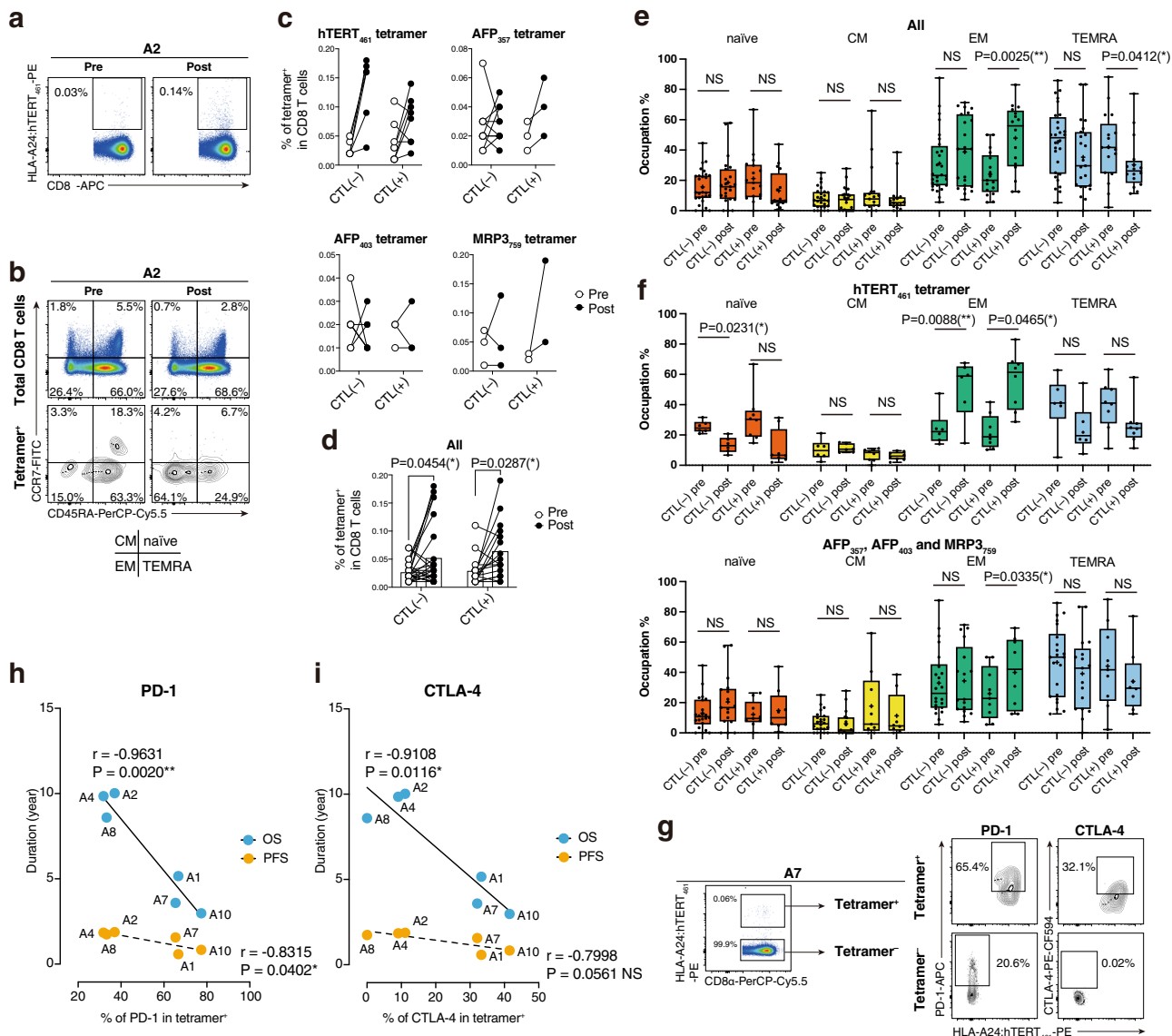

**Fig. 2 Induction of HLA-A24:peptide tetramer-binding T cells and phenotypic features.** A representative example of flow plots is shown. The frequency of the CD8α+tetramer+ fraction was calculated (**a**). Both total CD8+ T cells and tetramer+CD8+ T cells were detected by CD45RA and CCR7. Four fractions of naïve, central memory (CM), effector memory (EM), and terminally differentiated effector memory with CD45RA (TEMRA) are shown (**b**). Tetramer positivity among CD8+ T cells was compared between pre and postvaccination in CTL(−) and CTL(+) of each vaccine study. Samples from the same donors are connected by solid lines. **c** The results were combined in one graph. Samples from the same donors and tetramers are connected by solid lines. Mean values are shown by open bars. Two-way repeated-measures ANOVA followed by Bonferroni and Sidak's multiple comparison post hoc test was carried out. *P < 0.05. **d** Frequencies of naïve, CM, EM, and TEMRA fractions among tetramer+CD8+ T cells are displayed in box-and-whisker plots. CTL(−) (n = 26) and CTL(+) (n = 16) of pre and post are shown. Statistical analysis was conducted using two-way ANOVA followed by Bonferroni and Sidak's multiple comparison test to assess differences in composition among the different conditions of CTL induction and treatments. *P < 0.05; **P < 0.005; NS, not significant; error bar, maximum to minimum; horizontal lines of box, first quartile to third quartile; +, mean value (**e**). The frequencies of naïve, CM, EM, and TEMRA fractions among tetramer+CD8+ T cells of the hTERT study (n = 14) and the other three cohorts (n = 28) were analyzed separately. Two-way ANOVA followed by Bonferroni and Sidak's multiple comparison test was carried out. *P < 0.05; **P < 0.005; NS, not significant; whiskers, maximum to minimum; horizontal lines of box, first quartile to third quartile; +, mean value (**f**). Expression of inhibitory receptors PD-1 and CTLA-4 by HLA-A24:hTERT461 tetramer-binding CD8+ T cells was assessed. **g** The percentages of PD-1/CTLA-4 expression and overall survival (OS) period/progression-free survival (PFS) were plotted, then liner regression lines were calculated. r, Pearson's r; *P < 0.05; **P < 0.005; NS, not significant (**h**, **i**) Source data are provided as a Source Data file.

peptide-specific T cell functions. Peptide-specific TCRs were analyzed in two patients: A2 from the hTERT peptide vaccine study and D3 from the AFP peptide vaccine study. These patients both survived for more than 10 years after vaccination and exhibited the peptide-specific T cell inductions. As reported previously[13,14], we identified three hTERT461-specific TCRs in A2 and three AFP357-specific TCRs of D3 in the 1-year peripheral

blood sample (Fig. 4a, Supplementary Table 4). We tested these TCRs to confirm their Ag specificity and killing activities, which suggested that these TCRs played significant roles in controlling HCC development after peptide vaccination. hTERT-specific TCRs had a target cell killing activity only if the targets were loaded with the hTERT461 peptide (Fig. 4b). AFP-specific TCRs were evaluated in the same peptide-loaded cytotoxicity assay

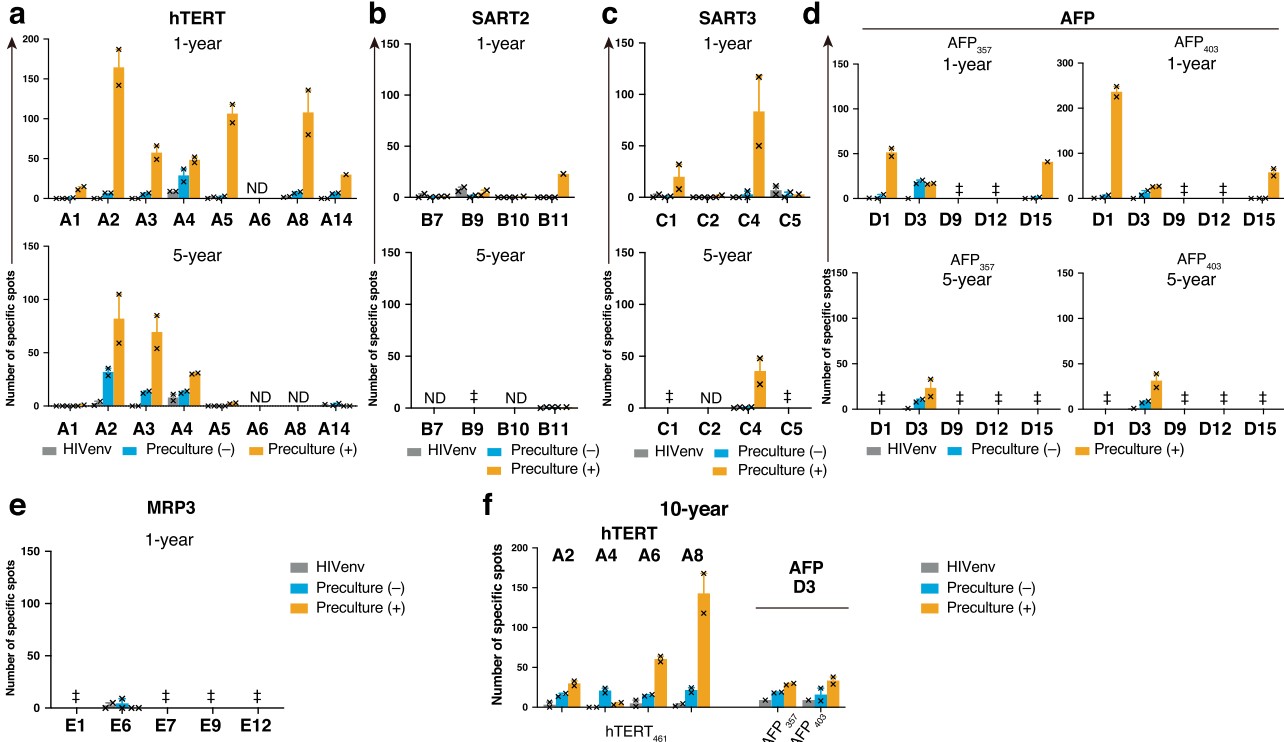

**Fig. 3 Persistent peptide-specific immune responses after vaccination.** PBMCs of CTL( + ) patients were collected at each timepoint and tested by the IFN-γ ELISpot assay. The experiments were performed in duplicate and mean numbers of specific spots are displayed in bar graphs per patient with experimental values (cross marks). The CTL( + ) cases among the hTERT participants were examined. The upper graph presents the 1-year result and the bottom graph shows the 5-year result. ND, not done because of unavailability of patient PBMC samples for a reason other than patient death. error bar, + s.e.m. **a** CTL( + ) cases of the SART2 peptide vaccine study are also shown. ‡PBMCs were unavailable because of patient death. ND, not done. bar, mean number of spots; error bar, + s.e.m. **b** Four patients with positive CTL induction in the SART3 trial were. ‡, PBMCs were unavailable because of patient death; ND, not done; bar, mean number of spots; error bar, + s.e.m. **c** For CTL( + ) participants in the AFP trial, ELISpot assays using AFP357 and AFP403 were separately carried out and shown in graphs. ‡PBMCs were unavailable because of patient death. bar, mean number of spots; error bar, + s.e.m. **d** ELISpot results of CTL( + ) patients from the MRP3 study are shown. Because no patient survived for 5 years, only one graph is shown. Because of the low cell number, a preculture experiment was not performed using E6 PBMCs. ‡PBMCs were unavailable because of patient death. ND, not done. bar, mean number of spots; error bar, + s.e.m. **e** At the 10-year timepoint after vaccination, there were only five patients who had survived: four patients in the hTERT cohort and one patient in the AFP study. These were all CTL( + ) cases. Immune responses against the peptides are shown in a graph. bar, mean number of spots; error bar, + s.e.m. **f** Source data are provided as a Source Data file.

system[11] and validated in the luciferase-based cytotoxicity assay. D3.14 and D3.16 exhibited significant cytotoxicities against endogenously processed AFP protein, whereas D3.2, which showed weaker cytotoxicity in the peptide-loaded system, did not act as a functional TCR (Fig. 4c). The killing activities of D3.14, A2.57, and A2.80 were visualized and confirmed by time-lapse imaging. These data demonstrated that peptide-specific TCR-transduced T cells showed cytotoxic actions after cell–cell contact with HepG2 cells, which resulted in more target cell death compared with control TCR (cytomegalovirus-specific TCR: see Supplementary Table 4)-transduced T cells (Fig. 4d–g).

**Peptide-specific T cells form a memory subset with a proliferative capability.** To further understand the characteristics of the peptide-specific T cells, we tracked the specific TCR chronotypes along with their transcriptome data over years after vaccination using the single-cell RNAseq technique. We again chose the 2 cases (A2 and D3) that showed more than 10 years of survival periods and we had already analyzed the peptide-induced TCR repertoires and functions both at 1 year and 5 years after vaccination.

To assess the transcriptional profiles of peptide-specific T cells in peripheral blood, we avoided in vitro expansion of Ag-specific T cells that were used for the previous TCR analysis because this

might have caused a critical bias in the results. Hence, we first checked tetramer positivity in CD8+ T cells of peripheral blood samples without stimulation. In the A2 donor at 1 year and 5 years after vaccination, hTERT-specific T cell frequencies were only 0.021% of CD8+ cells and 0.004% of CD8+ cells, respectively (Fig. 5a). D3 also showed quite a low population of AFP-specific T cells: 0.005% of CD8+ cells at 1 year and 0.006% of CD8+ cells at 5 years (Fig. 5b). This was perhaps due to the Ag-free environment in the body after vaccination was finished and tumors were eradicated. To analyze these limited numbers of cells, we enriched tetramer-positive cells of A2 and D3 by sorting (500 cells each) and mixed them with total lymphocytes from A2 (30,000) to prepare scRNA-seq samples (Supplementary Fig. 3).

We processed the cell suspensions to obtain transcriptome data at the single-cell level together with TCR sequence data as described in Methods, and then dissected the whole transcriptome information using the t-distributed stochastic neighbor embedding (t-SNE) dimension reduction method for 1- and 5-year samples. Both t-SNE plots generated about 10 clusters that roughly formed the major blood lymphocyte subsets including T cells, B cells, monocytes, dendritic cells, and NK cells (Fig. 5c, Supplementary Fig. 4). By integrating the TCR sequence data and scRNA-seq transcriptome data, we found known peptide-specific TCR sequences in the t-SNE plots. In the 1-year data, 4 hTERT-

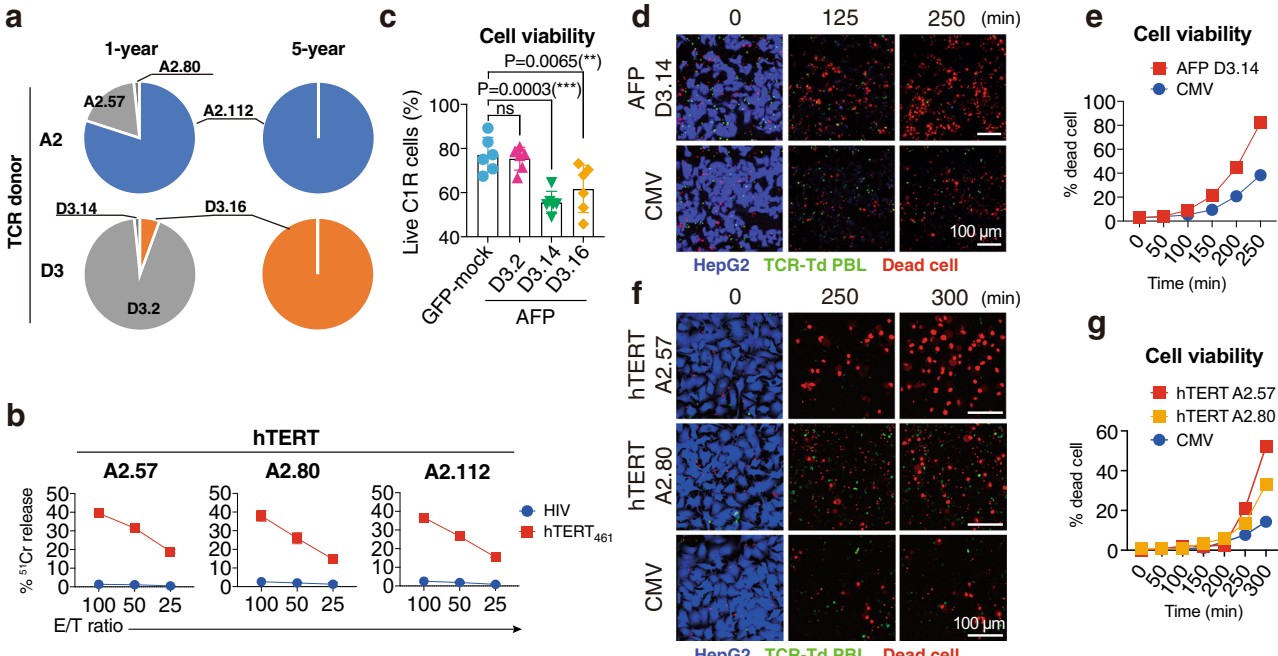

**Fig. 4 Peptide-specific T cell receptor repertoires over time.** A2 from the hTERT trial and D3 from the AFP trial were analyzed. AFP$_{357}$ was used to analyze the TCR repertoire in D3 in this experiment. The repertoires were presented in pie charts. The size of each pie indicates the frequency of the clone in the total obtained repertoire. The same pies and the same colors represent identical TCR pairs including TRAV/TRAJ and TRBV/TRBJ/TRBD as well as CDR3αβ sequences (**a**). Three TCR pairs obtained from A2 were tested for their function by $^{51}$Cr release assay. The experiments were done in triplicates. % $^{51}$Cr release were shown in the graphs. Error bar, $+/-$ s.e.m. **b**. Three TCR pairs obtained from D3 were tested by the luciferase-based killing assay. Mean viabilities from two independent experiments were combined and presented in a bar graph. Each experiment was performed with triplicates. One-way ANOVA was carried out followed by Bonferroni and Sidak's multiple comparison test. ***$P < 0.0005$; **$P < 0.005$; ns, not significant; error bar, $+/-$ s.e.m. **c** Cytotoxicity induced by the D3.14 of D3 is also shown using the HepG2 hepatoma cell line (HLA-A24$^+$AFP$^+$hTERT$^+$) and fluorescence microscopic time-lapse imaging. A CMVpp65$_{328}$-specific TCR was used as a negative control TCR (see Supplementary Table 4). Time-lapse snapshots are presented (0, 125, 250 min.). This experiment was duplicated and produced a similar result (Source data). **d** Percentages of dead cells were plotted on a graph over time (minutes) (**e**). Cytotoxicity induced by the A2.57 and A2.80 of A3 are tested as well using the HepG2 hepatoma cell line and the CMVpp65$_{328}$-specific TCR as a negative control. Time-lapse snapshots are presented (0, 250, 300 min.). **f** Percentages of dead cells were plotted on a graph over time (minutes) These experiments were duplicated and produced similar results as shown in the Source data. **g** Source data are provided as a Source Data file.

specific T cells out of 3,568 cells were identified, which comprised 3 different TCRs, A2.112, A2.80, and A2.57, as matched with the previous TCR repertoire analysis (Fig. 4a). However, no AFP-specific TCRs were found in this sample. In the 5-year data, four hTERT-specific T cells and five AFP-specific T cells were detected among 6,191 cells, which comprised A2.112, A2.80, D3.14, and D3.16, although the previous TCR repertoire analysis only found A2.112 and D3.16 at 5 years (Fig. 4a). This TCR information was embedded and shown as t-SNE plots, which revealed that the plotted locations of the hTERT-specific T cells of A2 were slightly out of the main area of CD8$^+$ T cells and indicated that these T cells may express different phenotypes from the other "regular" CD8$^+$ T cells. Additionally, the tetramer$^+$ T cells of D3 were projected close to the hTERT-specific T cells of A2 (Fig. 5d). To understand transcriptional signatures of the small population of the peptide-specific T cells, we determined the subcluster of these T cells. We focused on the $CD3E^+CD8A^+$ (CD8$^+$ T cell) population and reclustered using the t-SNE method followed by k-means clustering within this population, which resulted in three subclusters in both samples (Fig. 5e). Because most peptide-specific T cells belonged to cluster 2 of the 1-year sample and cluster 2 of the 5-year sample (Fig. 5f), we compared each with the other combined clusters. Sixteen genes (eight upregulated and eight downregulated genes) were identified as significantly differentially expressed genes in the 1-year sample (Fig. 5g). We also found 19 significantly differentially expressed genes in the

5-year sample (nine upregulated and 10 downregulated genes) (Fig. 5h). Pathway analyses of these genes revealed their transcriptional characteristics. The gene network of the 1-year sample shown in Supplementary Fig. 5 indicated downregulation of numerous cytotoxic proteins, including *GRZB*, *PRSS23*, and *FGFBP2*, while *GRZK* was highly upregulated. Glycolysis enzyme *HK3* and Src family tyrosine kinase *HCK* were also upregulated and a set of inhibitory membrane proteins with ITIM motifs were upregulated, including *LILRB2*, *SIGLEC7*, and *CLEC4A*. Five years after vaccination, the gene network shown in Supplementary Fig. 6 revealed a clearer molecular perspective. *IL7R* and *SELL* (CD62L) were obviously upregulated, which suggested a memory phenotype with durable survival periods. Upregulation of *NOSIP*, which antagonizes iNOS, might contribute to maintenance of the memory phenotype. The AP-1 transcription factor network, including *FOS*, *JUN*, and *JUNB*, might control manifestation of the memory phenotype. However, downregulation of CTSC, CST7, and HLA-DPs suggested low activity as effector CTLs. To compare the overall gene expression features of the peptide-specific T cells, we combined the scRNA-seq data of 1-year CD8$^+$ T cells and 5-year CD8$^+$ T cells that included peptide-specific T cells. We found accumulation of peptide-specific T cells in the t-SNE plot, although the clusters that contained these cells had rarely overlapped, which suggests that these cells shared transcriptomic features. On the other hand, CMV-specific T cells were located in the rest of the plot (Fig. 5e–f,

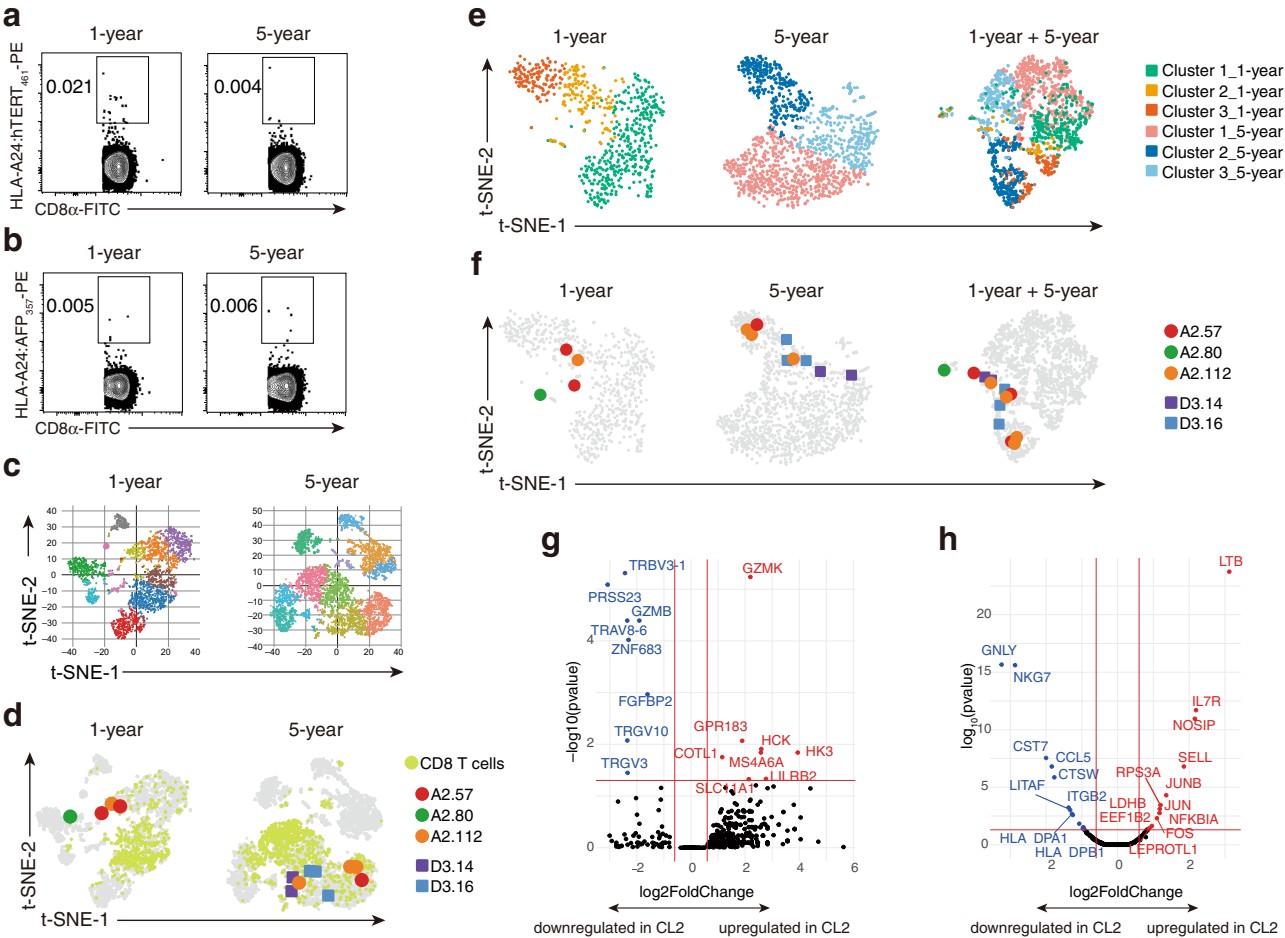

**Fig. 5 Transcriptome landscape of peptide-specific T cells.** Unstimulated PBMCs of patient A2 at each timepoint were stained with HLA-A24:hTERT$_{461}$ tetramer along with an anti-CD8α antibody and 7-AAD. Flow plots are presented with frequencies (%) of tetramer$^+$ cells among CD8$^+$ cells (**a**). Unstimulated PBMCs of patient D3 at each timepoint were stained with HLA-A24:AFP$_{357}$ tetramer along with the anti-CD8α antibody and 7-AAD. Flow plots are presented with frequencies (%) of tetramer$^+$ cells among CD8$^+$ cells (**b**). To proceed with such small frequencies of tetramer$^+$ cells, the 1-year tetramer$^+$ cells were enriched and mixed to prepare the 1-year enriched mixture. A 5-year enriched sample was also prepared. These samples were applied to the scRNA-seq analysis pipeline and the obtained transcriptome matrixes are presented in the t-SNE plots with clustering (see Supplementary Figure 3) (**c**). Single-cell VDJ-seq data were imported into the scRNAseq in accordance with their UMIs. Three hTERT-specific TCR sequences (A2.112, A2.80, and A2.57) and two AFP-specific TCR sequences (D3.14 and D3.16) were retrieved and visualized on the t-SNE plots. Cells that met the criteria of CD3E log2 fold change >0 and CD8A log2 fold change >0 are highlighted in light green (**d**). Cells that met the criteria of CD3E log2 fold change >0 and CD8A log2 fold change >0 were extracted and underwent the t-SNE dimension reduction followed by the k-means clustering again. Each plot generated three clusters (left and middle). The 1-year data and 5-year data were combined and subject to the t-SNE dimension reduction with the cluster marks held over (right) (**e**). Peptide-specific T cells were overlaid on the new plots (**f**). Differentially expressed genes were extracted by comparing the newly generated cluster (CL) 2 with the others of the 1-year t-SNE and then shown on a volcano plot (**g**). Differentially expressed genes were extracted by comparing newly generated cluster 2 with the others of the 5-year t-SNE and then shown on a volcano plot (**h**). Source data are provided as a Source Data file.

Supplementary Fig. 7). We first compared the expression of the gene set of interest. Both hTERT-specific T cells and AFP-specific T cells had upregulated inhibitory receptors including *PDCD1, HAVCR2, LAG3, KLRC1, CTLA4,* and *TIGIT.* Although they exhibited strong dysfunctional features, they expressed inflammatory cytokines *IFNG* and *TNF,* while *GZMB* expression was obviously lower. *CXCR5, IKZF2, TOX, CD27,* and *CD28* were upregulated in the peptide-specific T cells. *SELL* and *TCF7* were generally upregulated, while some exceptions such as D3.14 were observed (Supplementary Fig. 8). We next identified significantly differentially expressed genes in combined peptide-specific T cells, which included both hTERT$_{461}$ and AFP$_{357}$ detected in 1- or 5-year samples, compared with other CD8 T cells of 1- and 5-year samples. Four upregulated genes (*HIST1H3H, CCNA2, CDK1,* and *CXCL2*) and no downregulated genes were detected (Supplementary Fig. 9) suggesting active proliferation.

## Discussion

As immune checkpoint blockade therapeutics have been developed and become widespread, tumor immunity has been understood more profoundly. Exhaustion or dysfunction of T cells occurs in tumor microenvironments and during chronic viral infection, which show high expression of inhibitory molecules such as PD-1, Tim-3, and interleukin (IL)-10 and low production of proinflammatory cytokines including IFN-γ, TNF-α, and granzyme B, which lead to tumor progression or unsuccessful viral eradication[21,22]. However, there are also Tcf-1$^+$ stem-like memory T cells among PD-1$^+$ CD8$^+$ T cells, which form a long-lived subset that can proliferate and differentiate, especially under PD-1 blockade[23–25]. In this context, it is crucial to induce long-lived functional memory cells in developing immunotherapeutics with effective and durable antitumor effects.

In this post hoc observational study of the series of peptide vaccine clinical trials, we identified several 10-year-long-term survivors with IFN-γ-producing CTL induction. In these cases, we observed durable antitumor effects with persistently preserved immune responses by the same TCR clonotypes. However, the CTL induction rates were as low as one-third of the total, which can be improved by the administration of multiple peptides or a combination strategy with PD-1/PD-L1 blockade[26]. We found that an increase in the tetramer+ fraction of CD8+ T cells did not always indicate effective CTL induction, possibly because a peptide-MHC complex sometimes binds to a TCR without signal transduction due to inappropriate geometric topology between the two complexes[27] or the dysfunctional phenotype of CD8 T cells that fail to produce inflammatory cytokines upon antigen stimulation[28].

The peptide vaccines successfully introduced the formation of an effector memory phenotype. In the hTERT study, lower expression of PD-1 and CTLA-4 by hTERT$_{461}$-specific T cells after vaccination correlated with a better prognosis, which indicated that expression of these inhibitory receptors constrained the effector T cell functions. This finding highlighted a potential combination therapy with PD-1/PD-L1 blockade or CTLA-4 inhibition.

The vaccines maintained sustainable immune responses beyond 10 years without continuous administration of the peptides in selected patients. This has changed our understanding of cancer vaccine effects, namely that the immune responses induced by vaccines might be transient in most cases and disappear soon after vaccination[16]. The basic understanding of lifetime memory formation is that it occurs in the absence of antigen, such as after clearance of an acute viral infection[29]. During cancer with continuous antigen stimulation, CD8+ T cells enter a dysfunctional state marked by the expression of a series of inhibitory molecules and have difficulty in forming a memory phenotype that can control tumor progression[30]. PD-1/PD-L1 blockade can solve the problem of converting Tcf-1+PD-1+ CD8+ T cells to effector cells[31]. Peptide vaccines may also be another approach by directly inducing a memory phenotype in tumor-specific T cells.

To further understand the phenotypes of inducible CTLs by the peptide vaccines, we employed the single-cell RNA-seq method. In this study, the importance of the assay was to relate the transcriptome data to the TCR sequence data, so we could retrieve known TCR sequences with the transcriptome data simultaneously. The same strategy has also been adopted by others to analyze tumor-infiltrating T cells and characterize dysfunctional T cells[32]. The novelty of our approach was that we already had the TCR repertoire dataset with specificity confirmation and functional validation before obtaining the scRNA-seq data. Under cancerous conditions, because there are abundant clonally expanded CD8+ T cells that are not specific to tumors, which are named bystander CD8+ T cells[33], our method was the only strategy to clarify the phenotypes of truly tumor-specific CD8+ T cells.

We successfully identified peptide-specific CD8+ T cells at 1 and 5 years after vaccination, although the number was quite low even after enrichment. In the AFP D3 patient, we found two cells that expressed TCR D3.14 that had the highest avidity among the AFP$_{357}$-specific TCRs and had been only seen in one cell before[11,14]. We observed a dysfunctional phenotype with increased inhibitory molecules and a stem-like phenotype with TCF7 and CXCR5 expression simultaneously in peptide-specific CD8+ T cells, which suggested potential memory formation during the long-term periods. Next, we performed comprehensive gene expression analysis followed by pathway analysis. The 1-year data showed upregulation of inhibitory membrane molecules

with the ITIM motif together with a Src family tyrosine kinase, which might downregulate effector cytokines. The 5-year data of peptide-specific CD8+ T cells revealed the typical memory phenotype or stem cell-like properties with IL7R and SELL expression, and upregulation of the AP-1 transcription family including JUN, JUNB, and FOS, which might contribute to resistance against T cell exhaustion[34,35]. These cells also upregulated NOSIP expression, which also led to memory formation[36]. These findings of phenotypic changes over time after vaccination suggested that the memory formation occurred gradually over long-term periods. Our 5-yar data indicated active proliferation with increased expression of AP-1 transcription factors, which may contribute to the memory formation of the antigen-specific CD8+ T cells[37] that were detectable by the IFN-γ ELISpot assay even at 10 years after vaccination. These T cells were also characterized more directly by the mitosis driving genes including CCNA2, CDK1 and HIST1H3H suggesting relentless proliferation and self-renewal instead of quiescent long-lived T cells. The durable T cell function specific for cancer also suggests possible application to TCR-T cell therapies using these TCR genes to overcome the disadvantage of the low CTL induction rate[38]. We note that our scRNA-seq results were based on the limited samples that only included two patients due to sample availability.

In conclusion, peptide-specific CD8+ T cells were maintained beyond 10 years after peptide vaccination. The results suggest that induction of memory T cells with a self-renewal capability is critical for durable antitumor immune responses.

## Methods

**Peptide vaccine trials and patients**. We reanalyzed 65 HLA-A24-positive HCC patients who were enrolled in hTERT-derived peptide vaccine (UMIN000003511), SART2-derived peptide vaccine (UMIN000004540), SART3-derived peptide vaccine (UMIN000005677), AFP-derived peptide (UMIN000003514), or MRP3-derived peptide (UMIN000005678) phase I trials. Each patient was histologically or radiologically diagnosed with primary HCC in accordance with the American Association for the Study of Liver Diseases guidelines for the management of HCC. hTERT, SART2, and SART3 trials were designed for the development of adjuvant therapeutics after local radiofrequency thermal ablation therapy for early-stage HCC. The AFP vaccine was administrated as a monotherapy for advanced HCC. The MRP3 vaccine was a combination therapy with hepatic arterial infusion chemotherapy of 5-FU and cisplatin for advanced HCC. The inclusion criteria were a Karnofsky score of ≥70%, ≥20 years of age, informed consent, and the following normal baseline hematological parameters: ≥8.5 g/dl hemoglobin; ≥2000/μl white blood cell count; ≥50,000/μl platelet count; ≤1.5 mg/dl creatinine, and Child-Pugh classification A or B. The exclusion criteria were severe cardiac, renal, pulmonary, hematological, or other systemic diseases associated with a discontinuation risk. Such diseases included human immunodeficiency virus (HIV) infection, prior history of other malignancies, the recent history of surgery, chemotherapy or radiation therapy within 4 weeks before registration, immunological disorders including splenectomy and radiation to the spleen, corticosteroid or anti-histamine therapies, currently lactating or pregnant, history of organ transplantation, and difficulty in follow-up. Informed consent was obtained in each patient. We did not pay for participation in the studies. The primary endpoints were the evaluation of the safety and immunological effects of each peptide vaccine. The secondary endpoints were the evaluation of the recurrence rate and tumor markers in the studies of hTERT, SART2, and SART3, and anti-tumor effects in the studies of AFP and MRP3. The treatment protocol of each study has been described previously. Briefly, patients received 0.03−3.0 mg hTERT (hTERT$_{461}$), SART2 (SART2$_{899}$), SART3 (SART3$_{109}$), AFP (AFP$_{357}$ and AFP$_{403}$), or MRP3 (MRP3$_{765}$)-derived peptide vaccines. The dose of hTERT$_{461}$, SART2$_{899}$, SART3$_{109}$, and MRP3$_{765}$ peptides was increased from 0.03 to 3.0 mg until dose-limiting toxicity was observed. In the AFP trial, 3 mg each of AFP$_{357}$ and AFP$_{403}$ peptides were injected and the vaccinations were continued until the patients showed progressive disease. The peptide was emulsified in incomplete Freund's adjuvant (Montanide ISA-51 VG; SEPPIC, Paris, France) and administered by subcutaneous immunization 3–57 times every other week. After treatment, HCC recurrence was evaluated by dynamic CT or MRI every 3 months for 30 months in groups A–C. In group D and E, dynamic CT or MRI were performed to evaluate the anti-tumor effect every 2–3 months. All patients had provided written informed consent to participate in the study. The study protocols and the use of the clinical samples for this study conformed to the ethical guidelines of the 1975 Declaration of Helsinki and were approved by the regional ethics committee (Medical Ethics Committee of Kanazawa University) (trial registration: UMIN000003511, UMIN000004540, UMIN000005677, UMIN000003514, UMIN000005678, and 2016-122).

**Peptides and preparation of PBMCs.** Detailed information on each peptide, including the amino acid sequence, is provided in Supplementary Table 1. To immunize HCC patients with peptide vaccines, we synthesized hTERT (hTERT$_{461}$), SART2 (SART2$_{899}$), SART3 (SART3$_{109}$), AFP (AFP$_{357}$ and AFP$_{403}$), and MRP3 (MRP3$_{765}$)-derived peptide as GMP grade at Neo MPS, Inc. (San Diego, CA, USA). For immunological analyses, hTERT$_{461}$, SART2$_{899}$, SART3$_{109}$, AFP$_{357}$, AFP$_{403}$, and MRP3$_{765}$ peptides, HIV envelope-derived peptide (HIVenv$_{584}$), and CMV pp65-derived peptide (CMVpp65$_{328}$) were purchased from Sumitomo Pharmaceuticals (Osaka, Japan). They were validated by mass spectrometry and HPLC with > 90% purity. PBMCs were isolated before and every 4 weeks after one course of peptide vaccine (three injections) as described previously[10–13]. PBMCs were resuspended in RPMI 1640 medium containing 80% FCS and 10% dimethyl sulfoxide, and cryopreserved until use.

**IFN-γ ELISpot assay.** IFN-γ ELISpot assays were performed as reported previously[13]. The negative control was an HIV envelope-derived peptide (HIVenv$_{584}$). Positive controls were 10 ng/ml phorbol 12-myristate 13-acetate (Sigma) or a CMV pp65-derived peptide (CMVpp65$_{328}$). Colored spots were counted by a KS ELISpot Reader (Zeiss, Tokyo, Japan). The number of specific spots was determined by subtracting the number of spots in the absence of an antigen from the number in the test well. Positive cytotoxic T cell (CTL) induction, CTL(+), was defined as a ≥10 specific spots increase and two-fold increase. Under the preculture condition, PBMCs were cultured in the presence of the antigen peptide (1 µg/ml) and IL-2 (50 U/ml, Shionogi & Co., Ltd., Osaka, Japan) before the ELISpot assay. For immune monitoring, PBMCs were not only tested directly (preculture−) but also examined after in vitro peptide stimulation (preculture +) for 7 days. The prepared PBMCs were incubated with or without the peptide that had been used in the clinical trial or HIV-gp160$_{584}$ peptide as a control.

**Cell lines.** C1R-A24 cells were maintained in RPMI-1640 medium (Wako Pure Chemical Industries Ltd., Osaka, Japan) containing 10% fetal bovine serum (Thermo Fisher Scientific, Inc. Waltham, MA, USA), 100 µg/mL streptomycin, 100 U/mL penicillin, and 500 µg/mL hygromycin B[14]. HepG2 and Phoenix-A were maintained in Dulbecco's modified Eagle's medium (Wako Pure Chemical Industries Ltd.) containing 10% fetal bovine serum, 100 µg/mL streptomycin, and 100 U/mL penicillin. C1R-A24-AFP-Luc cells were generated by retrovirally transducing AFP cDNA and luciferase expression vectors into C1R-A24 cells[11].

**Flow cytometry.** The antibodies used in this study are listed in Supplementary Table 5. The antibodies were used in 1:100 dilution except SFCl21Thy2D3 with 1:50 dilution. Flow cytometry and cell sorting were conducted on a BD FACSAria II. Raw data were collected by BD FACS Diva 6.1.2 software that exported FCS files. To analyze the FCS files, we used FlowJo ver. 10.7.1 (BD) to obtain the frequencies of a certain population and generate figure components.

**Induction of TAA-specific T cells and TCR cloning.** PBMCs were collected from patients A2 and D3 vaccinated with hTERT and AFP-derived peptides, respectively. The collected PBMCs were cultured in the presence of 10 µg/mL hTERT$_{461}$ or AFP$_{357}$ peptide, 10 ng/mL recombinant interleukin (rIL)−7, and 100 pg/mL rIL-12 (PeproTech, Inc., Rocky Hill, NJ, USA) in a 96-well U-bottom plate. On days 3, 10, and 17, half of the culture medium was replaced with 350 U/mL rIL-2 (PeproTech, Inc.)-containing medium. On days 7 and 14, half of the culture was replaced with medium containing 350 U/mL rIL-2, 20 µg/mL peptide, and mitomycin-C-treated autologous PBMCs. After 3 weeks of in vitro culture, the generated CTLs were stained with phycoerythrin-conjugated peptide MHC tetramers (MBL Co., Ltd., Aichi, Japan) after Fc receptor blocking (Clear Back; MBL Co., Ltd.), followed by a fluorescein isothiocyanate-conjugated anti-CD8α antibody (MBL Co., Ltd.) and 7-AAD (Beckman Coulter, Inc., Indianapolis, IN, USA) staining. Ag-specific CD8$^+$ T cells were detected and collected as single cells using a FACSAria II (BD Bioscience, San Diego, CA, USA). cDNAs of TCR α and β chains were amplified by the 5′ rapid amplification of cDNA end method and reverse transcription-polymerase chain reaction from single cells. cDNAs were analyzed with repertoires after sequence analyses and confirmation of Ag specificities using an hTEC10 system as described previously[14].

**TCR transduction into PBMCs.** TCR β and α chains were linked via a viral P2A sequence, cloned into a pMXs-IRES-GFP vector, and transfected into Phoenix-A retroviral packaging cells. Each TCR constant region was optimized for codons. Viral supernatant was filtered and applied to plates coated with 50 µg/mL Retro-Nectin (Takara Bio Inc., Shiga, Japan) and spin-loaded by centrifugation. PBMCs were stimulated in vitro for 2 days with CD3/CD28 beads (Thermo Fisher Scientific) and 30 U/mL rIL-2 prior to infection, added to the plate, and infected with the retrovirus. Retroviral infection was repeated twice. TCR-transduced PBMCs were used in assays at 10–14 days after the start of stimulation.

**Cell killing assays.** In the $^{51}$Cr release assay, $5 \times 10^5$ C1R-A24 cells loaded with 10 µg/mL hTERT$_{461}$ peptide or HIVenv$_{584}$ peptide (MBL Co., Ltd.) overnight were labeled with 0.925 MBq $^{51}$Cr and then cocultured with $1.2 \times 10^5$ K562 cells and hTERT TCR-transduced PBMCs at ratios of 100:1, 50:1, or 25:1 for 4 h. $^{51}$Cr released into the supernatant was counted. %Specific lysis was calculated by the formula: (experimental release (cpm) − spontaneous release (cpm))/(maximum release (cpm) − spontaneous release (cpm)) × 100. The luciferase-based killing assay was performed by coculturing $1 \times 10^3$ C1R-A24-AFP-Luc cells (and $5 \times 10^4$ K562 cells for 24 h with AFP TCR-transduced PBMCs at a ratio of 50:1. Target cell viability was determined by measuring luciferase expression using luciferase assay substrate (Promega corp., Fitchburg, WI, USA). Luciferase activity of target cells incubated without effectors was considered 100%. Cytotoxicity was also evaluated by time-lapse imaging of dead and live cells in the coculture system. HepG2 cells spread onto an 8-well chambered glass slide on the preceding day were stained with 10 µM calcein violet (Thermo Fisher Scientific) for 30 min at 37 °C, cocultured with $3 \times 10^5$ EGFP-expressing TCR-transduced T cells, and then observed under a confocal microscope in the presence of propidium iodide. Percentages of dead cells were calculated by the formula %dead cell = (dead cell number)/(initial HepG2 cell number) × 100).

**Single-cell analysis of T cells.** Single-cell RNA-sequencing (scRNA-seq) and single-cell TCR-sequencing (scTCR-seq) were performed simultaneously using Chromium single-cell gene expression solution and Cell Ranger 4.0 Analysis Pipelines (10x Genomics, Pleasanton, CA USA)[39]. A cell suspension was mixed with barcoded gel beads and placed in oil to prepare a gel beads-in-emulsion in which cDNAs from a single cell were attached to the same barcode in droplets by the chromium unit. Both gene expression and TCR libraries were generated and sequenced from the cDNA. Sequence reads were processed by Cell Ranger Analysis Pipelines to output files that were readable by Loupe browser 4.0/5.0 (10x Genomics). The TCR sequence data were linked to the gene expression data in the Loupe browser and then t-SNE plots and gene expression profiles were generated with statistical values of each subset. Heatmaps and volcano graphs were generated by R software. Volcano plots generate red dots and gene symbols in red are significantly upregulated genes as judged by $P < 0.05$ and log2 fold change > 1 and blue dots and gene symbols in blue are significantly downregulated genes as judged by $P < 0.05$ and log2 fold change <−1. Pathway analysis was conducted on the Cytoscape and GeneMANIA prediction server: biological network integration for gene prioritization and predicting gene function[40] (Supplementary Note).

**Statistical considerations and graph generation.** Statistical analyses, which included the Fisher's test, one-way ANOVA, two-way repeated-measures ANOVA, Bonferroni and Sidak's multiple comparison post hoc test, correlation ecoefficiencies using Pearson's r formula, and the related graph generation, were performed using Prism v_9.0.0 & v_9.1.0 (GraphPad software, San Diego, CA, USA) or R software. For the differential gene expression analysis that tested mean expression of genes between clusters in the scRNA-seq experiment, the exact negative binomial test or fast asymptotic negative binomial test was performed to generate p-values adjusted for multiple testing using the Benjamini-Hochberg procedure to control FDR [https://github.com/10XGenomics/cellranger/blob/master/lib/python/cellranger/analysis/diffexp.py].

**Reporting summary.** Further information on research design is available in the Nature Research Reporting Summary linked to this article.

## Data availability
Source data are provided with this paper. The RNAseq data generated in this study have been deposited in the DNA data bank of Japan (DDBJ) under the bioproject number PRJDB11756. The raw RNAseq data are available in the depository. The processed data are available in the Source data file.

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

## Acknowledgements

This project was supported by research grants from the Ministry of Education, Culture, Sports, Science, and Technology of Japan (18H02794) and AMED under grant numbers JP20fk0210077 and JP19fk0310116. This work was also funded in part by the Industry-Academia Collaboration Courses, supported by Eisai Co., Ltd. We thank Mitchell Arico from Edanz Group (https://en-author-services.edanz.com/ac) for editing a draft of this manuscript.

## Author contributions

E.M, H.N., and S.K. designed the research; E.M., H.N., Tatsuya. Y. collected the clinical data and samples; E.M., H.N., K.F., K.N. K.A. and Taro. Y. managed the clinical data and samples; E.M., H.N., and K.F. performed the experiments; E.M., H.N., and K.F. analyzed the data; E.M., H.N., T. Tamai., M.K., T. Terashima, N.I., K.A., Tatsuya. Y., Taro. Y., Y.S., and M.H. discussed the data; and E.M. and H.N. wrote the manuscript.

## Competing interests

The authors declare no competing interests.
