## [Peer Review File · Nature Communications]

Peptide vaccine-treated, long-term surviving cancer patients harbor self-renewing tumor-specific CD8⁺ T cellsREVIEWER COMMENTS

Reviewer #1 (Remarks to the Author):

In this study, Mizukoshi et al. tracked tumor peptide specific TCR clonotypes at single cell level in hepatocellular carcinoma patients challenged with cancer vaccine for more than 10 years. They revealed that particular peptide specific TCR clones of CD8+ T cells were maintained for ten years acting as self-renewal memory phenotype to provide effective anti-tumor immunity. This study provides critical findings on stem-like feature of anti-cancer specific T cells and possible application for TCR-T cell therapy in cancer patients. I have no major issue with this manuscript.

Minor comment:

Figure 2a, it is difficult to distinguish hTERT group from MRP3 group.

Reviewer #2 (Remarks to the Author):

Mizukoshi E. et al performed a detailed characterization of peptide-specific CD8+ T cell responses in patients with HCC treated with vaccines targeting distinct tumor antigens. Further, they detect TCRs specific for hTERT and AFP and track some of these TCRs at single cell level to perform transcriptomic analysis at 1 and 5 years after vaccination. They detect transcriptomic changes over time in the peptide specific TCR clonotypes from a less cytotoxic and more exhausted state to more typical memory phenotype expressing IL7R and SELL.

Whilst the data is highly interesting and relevant for the field of cancer immunology and personalized cancer vaccines, the data present some weaknesses. Some are inherent to the low frequency of peptide-specific responses in peripheral blood, but they raise some questions regarding the conclusions.

Major points:

1. Responses against control peptides CMV pp65 also increase in post-treatment samples compared to pre-treatment (example Fig. 1a and b). Is the improved outcome due to the induction of specific immune response against vaccine or an improvement in the general functionality of T cells, regardless of the specificity?

In Figure 1c and d, how would the CMV responses look in these patients? Would CMV pp65 CTL induction be associated with better clinical outcome?

2. Could other effector functions other than IFN-gamma secretion explain discrepancies between the % tetramer and % CTL induction?

3. It would be of interest to provide methodological details about the sorting and mixing of CD8+ cells and PBMCs prior to scRNA sequencing. Were the PBMCs from A2 patient with which the tetramer+ cells from A2 and D3 patients were mixed also sorted? Could this account for transcriptomic differences between the sorted and unsorted cells?

4. The fact that the tetramer+ cells from D3 were sorted and mixed with PBMCs of A2 is confounding, as it is unclear what the reference transcriptomic traits of CD8+ T cells in this patient would look like.

5. Please clarify, do the A2 derived hTERT-specific TCRs recognize the peptide naturally processed and presented by tumor cells?

6. The main limitation of the study is that the events or number of patients studied and the number of peptide specific TCR clonotypes detected in the single cell transcriptomic analysis is scarce. In addition, the exact same clonotype is not always detected in the pre and post-treatment sample. This limits the ability to conclude that these are the same exact cells that have changed transcriptomically.

7. Given that the conclusion is that vaccines can generate such long term memory responses, it would be of interest to increase the number of patients studied and track the transcriptomic profile of CMV specific cells at the same time points to see if these changes are unique to vaccine specific TCRs, but not CMV-specific lymphocytes.

8. In Figure 6d and e, it would be preferable to provide clustering for both 1 year and 5 year all together instead of providing separately so clusters and TCRs detected in specific clusters can be directly compared.

Minor points:

1. Figure legend 1d is missing

2. Figure 3g FMO control for Tet+ and Tet- could be included in the Supplementary data to ensure that the expression of PD-1 reported has been well compensated.

3. Lines 215-217, please clarify which peptides specific cells are being compared.

4. Line 258: add "in selected patients"

5. Lines 275 and 276 differences between finding two cells and one cell at different time points does not support that this TCR was involved in controlling disease.

6. Technical details flow cytometry, including compensation and how gating was done strategy and negative controls (specific use of FMO controls, if they were used).

7. Did the authors analyze the presence of responses in CTL- patients as well as negative control?

Reviewer #3 (Remarks to the Author):

The statistical methods applied by Mizukoshi et al. in this manuscript are appropriate and support the conclusions of the study. Sample sizes are admittedly modest, however considering the nature of the study I support that the analysis of the available data is suitable and the biological significance of the findings should be evaluated despite modest sample sizes. The authors have performed survival analysis using log ranked test, reporting p-values and hazard ratios. I would recommend the authors to perform multivariate survival analysis taking into account the traditional predictors of outcome and incorporate the results in the manuscript. The authors should mention the test used for the differential gene expression analysis in the figure legends (Figure 6 and the associated supplemental figures) and in the Methods. Also, in the same analyses the p-values don't seem to have been adjusted for multiple testing and may not withstand multiple testing correction but I support that this is still valuable information of the differences in the comparison groups and suitable for publication. In supplementary figure 5 and 6 the range of the p-values should be indicated as graphical key or in the legend text. The authors have appropriately provided details of the other tests used in the relevant figure legends and there is a statement in the Methods mentioning the software used to generate the statistics.

RESPONSES to REVIEWER COMMENTS

We thank the reviewers for their valuable comments and indications. We have amended and strengthened our manuscript thanks to these valuable remarks. We have responded to the reviewers' comments on a point-by-point basis.

Reviewer #1 (Remarks to the Author):

In this study, Mizukoshi et al. tracked tumor peptide specific TCR clonotypes at single cell level in hepatocellular carcinoma patients challenged with cancer vaccine for more than 10 years. They revealed that particular peptide specific TCR clones of CD8+ T cells were maintained for ten years acting as self-renewal memory phenotype to provide effective anti-tumor immunity. This study provides critical findings on stem-like feature of anti-cancer specific T cells and possible application for TCR-T cell therapy in cancer patients. I have no major issue with this manuscript.

Minor comment:

Figure 2a, it is difficult to distinguish hTERT group from MRP3 group.

Response: We appreciate this comment to improve the presentation of the figure. We have changed the graph colors to a universal color combination (Fig. 2).

Reviewer #2 (Remarks to the Author):

Mizukoshi E. et al performed a detailed characterization of peptide-specific CD8+ T cell responses in patients with HCC treated with vaccines targeting distinct tumor antigens. Further, they detect TCRs specific for hTERT and AFP and track some of these TCRs at single cell level to perform transcriptomic analysis at 1 and 5 years after vaccination. They detect transcriptomic changes over time in the peptide specific TCR clonotypes from a less cytotoxic and more exhausted state to more typical memory phenotype expressing IL7R and SELL.

Whilst the data is highly interesting and relevant for the field of cancer immunology and personalized cancer vaccines, the data present some weaknesses. Some are inherent to the low frequency of peptide-specific responses in peripheral blood, but they raise some questions regarding the conclusions.

Major points:

1. Responses against control peptides CMV pp65 also increase in post-treatment samples compared to pre-treatment (example Fig. 1a and b). Is the improved outcome due to the induction of specific immune response against vaccine or an improvement in the general functionality of T cells, regardless of the specificity?

In Figure 1c and d, how would the CMV responses look in these patients? Would CMV pp65 CTL induction be associated with better clinical outcome?

Response: As the reviewer indicates, it would be interesting to determine whether irrelevant immune induction occurs after the peptide vaccine and is associated with

the outcome. We assessed CMV-specific immune induction using the exact definition of CTL induction in this manuscript. CMV-specific immune induction was seen in one out of the 14 hTERT patients, 0 out of the 12 SART2 patients, two out of the 12 SART3 patients, two out of the 15 AFP patients, and four out of the 12 MRP3 patients. The induction rate was generally lower than peptide-specific CTL induction (Supplementary Fig. 1). We next examined whether the CMV-specific immune induction was associated with overall survival. We performed the log-rank test that compared survival periods with CMV induction (+) and (–), which resulted in no differences. We further tested whether CMV immune induction correlated with the peptide-specific CTL induction using 2×2 contingency table analyses with Fisher’s exact test. The results showed no statistical correlation between these two immune inductions in any of the vaccine trials (Supplementary Table 4). Therefore, we concluded that the CMV immune inductions did not indicate a favorable outcome or accompany the peptide vaccine-specific immune inductions. We have added the text below:

Lines 114–115

“whereas irrelevant immune induction of CMV-specific responses did not correlate with the CTL induction or outcome (Supplementary Fig. 1, Supplementary Table 4).”

2. Could other effector functions other than IFN-gamma secretion explain discrepancies between the % tetramer and % CTL induction?

Response: We analyzed cytokine secretion and tetramer-binding capacity using A2 PBMCs collected 10 months after vaccination. The PBMCs were cultured in the presence of 1 $\mu\text{g}/\text{ml}$ peptide (HIVenv or hTERT₄₆₁) and protein transporter inhibitor (monensin) for 5 h. IFN γ and TNF were stained intracellularly after fixation for surface staining that included tetramers. We found 0.061% tetramer⁺ cells among CD8⁺ T cells, which was sufficient for analysis. First, we examined proinflammatory cytokine secretion by CD8 T cells and found an increased IFN γ ⁺ fraction (0.026%) and TNF⁺ fraction (0.026%) as well as double positives (0.013%) among CD8 T cells when cultured with the hTERT peptide compared with the HIVenv peptide. Notably, there were numerous cells that only secreted TNF, but not IFN γ . These cells might have been detected by tetramers, but not by the IFN γ ELISpot. This could be a reason for the discrepancy as the reviewer implied (see below).

Cytokine production by A2 CD8 T cells in response to hTERT peptide A2 PBMCs were incubated in the presence of the peptide and monensin for 5 h and then stained intracellularly to detect interferon- γ and TNF.

We have also reported another phenomenon. We observed slightly different TCR repertoires when we examined and compared tetramer⁺ and IFN γ ⁺ cells. We found some TCRs only in IFN γ ⁺ cells, but not in the tetramer⁺ cells, while 86% of the TCRs identified in IFN γ ⁺ cells were shared with the tetramer⁺ cells. Surprisingly, all TCRs unique to IFN γ ⁺ cells bound the tetramer in the *in vitro* settings (Kobayashi, Mizukoshi et al. Nat Med 2013). This finding indicated that we could not detect specific T cells

that produce IFN γ in response to the antigens when we used tetramers, which seemingly contributed to the discrepancies. Additionally, other studies have reported that the geometric distance between TCR and MHC peptide is critical for signal transduction to activate T cells. Thus, there might be non-functional bindings of TCR and pMHC (or tetramers) (Adams et al. *Immunity* 2011).

T cell dysfunction/exhaustion is another mechanism that generates unreactive CD8 T cells. It has been reported that PD-1KO Ag-specific T cells proliferate and produce more IFN γ than WT T cells, even when they are self-reactive (Keir et al. *J Immunol* 2007). We also observed similar findings. Using A2 PBMCs, we found a dramatic reduction of the tetramer⁺ fraction upon cognate peptide stimulation, which was due to TCR downregulation (Cai et al. *J. Exp. Med.* 1997) or trogocytosis (Hamieh et al. *Nature* 2019). The remaining tetramer⁺ cells after stimulation showed enrichment of PD-1⁺ cells (these cells were negative for IFN γ or TNF), which suggested PD-1⁺ Ag-specific T cells were unresponsive. This might explain why A7 with the high PD-1 in tetramer⁺ fraction had a negative ELISpot and poor prognosis (presented later in response to Reviewer #2's minor comment #7) and why PD-1 expression in tetramer⁺ cells correlated with prognosis (see Fig. 3h).

There might be multiple factors that underlie the discrepancies between cytokine production and tetramer positivity. Pursuing this issue is beyond the scope of this

manuscript, but may lead to a profound and essential understanding of the immune responses in cancer.

Reduction of the tetramer+ fraction of A2 CD8 T cells in response to hTERT peptide A2 PBMCs incubated in the presence of the peptide and monensin for 5 h were also stained with the hTERT tetramer and anti-PD-1 Ab.

We have added the text below:

Lines 255–256

“or the dysfunctional phenotype of CD8 T cells that fail to produce inflammatory cytokines upon antigen stimulation²⁹.”

3. It would be of interest to provide methodological details about the sorting and mixing of CD8⁺ cells and PBMCs prior to scRNA sequencing. Were the PBMCs from A2 patient with which the tetramer⁺ cells from A2 and D3 patients were mixed also sorted? Could this account for transcriptomic differences between the sorted and unsorted cells?

Response: We have provided the detailed method for single cell analyses. The sorting strategy was as follows: 30,000 cells of the 7-AAD⁻ fraction (Supplementary Fig. 2b) of donor A2, 1,000 hTERT₄₆₁ tetramer⁺ cells of donor A2, and 1,000 AFP₃₅₇ tetramer⁺ cells of donor D3 were bulk sorted using the FACSaria II purity mode and then combined

together. Then, 20,000 cells of the mixed cells were used for cDNA preparation by the Chromium system (10x Genomics).

The sorting results are shown in Supplementary Fig. 3. Because we sorted both tetramer⁺ and total lymphocytes, we assumed there was no transcriptomic bias associated with sorting between tetramer⁺ cells and total lymphocytes.

We have added a reference to *Supplementary Fig. 3* on line 193.

4. The fact that the tetramer⁺ cells from D3 were sorted and mixed with PBMCs of A2 is confounding, as it is unclear what the reference transcriptomic traits of CD8⁺ T cells in this patient would look like.

Response: We appreciate the reviewer's concern about the characterization of the tetramer⁺ cells from D3 without its reference transcriptome. We needed to explain the concept of the experiment more because we used a very complicated strategy for this assay. Because we did not analyze background transcriptome profiles of D3, we cannot discuss the tetramer⁺ cells of D3 in the context of their distinction from other CD8 T cells as noted by the reviewer. This analysis showed tetramer⁺ cells from D3 biasedly plotted in the A2 CD8 T cell field, which was the result of multiple transcriptomic comparisons between D3 tetramer⁺ cells and A2 CD8 T cells. This analysis did not address how the tetramer⁺ cells of D3 were characterized or functioned differently from the other CD8 T cells of D3. However, we would like to emphasize that the tetramer⁺ cells of D3 were more closely plotted to the tetramer⁺ cells of A2 compared with many of the A2 background CD8 T cells. This is very

interesting because two different peptide-specific T cells in different patients with different disease conditions shared the same transcriptomic features. As the reviewer mentioned, we should clarify this point. We have changed the text below:

Lines 205–208

“which revealed that the plotted locations of the hTERT-specific T cells of A2 were slightly out of the main area of CD8⁺ T cells and indicated that these T cells may express different phenotypes from the other “regular” CD8⁺ T cells. Additionally, the tetramer⁺ T cells of D3 were projected close to the hTERT-specific T cells of A2 (Fig. 6d).”

5. Please clarify, do the A2 derived hTERT-specific TCRs recognize the peptide naturally processed and presented by tumor cells?

Response: We appreciate the opportunity to introduce our new data here. We are currently focusing on phenotyping the functionality of each TCR, so that we can use these TCRs for therapeutic applications that include TCR-T therapies. We are determining whether the obtained TCRs recognize naturally processed antigens expressed by tumor cells. Thus far, we have seen that AFP-specific and hTERT-specific TCRs recognize and kill HepG2 hepatoma cells that are positive for HLA-A24, AFP, and hTERT. We have already shown the AFP D3.14 result in Fig. 5d, e. Additionally, we have added our new data on hTERT A2.57 and A2.80 in Fig. 5 f, g.

We have amended Fig. 5 and changed the text below in the manuscript:

Lines 174–178

*“The killing activities of D3.14, A2.57, and A2.80 were visualized and confirmed by time-lapse imaging. These data demonstrated that peptide-specific TCR-transduced T cells showed cytotoxic actions after cell–cell contact with HepG2 cells, which resulted in more target cell death compared with control TCR (cytomegalovirus-specific TCR: see **Supplementary Table 6**)-transduced T cells (Fig. 5d–f).”*

6. The main limitation of the study is that the events or number of patients studied and the number of peptide specific TCR clonotypes detected in the single cell transcriptomic analysis is scarce. In addition, the exact same clonotype is not always detected in the pre and post-treatment sample. This limits the ability to conclude that these are the same exact cells that have changed transcriptomically.

Response: We have had difficulty in analyzing the small number of Ag-specific T cells detected in FACS and the scRNA-seq datasets in this study. As the reviewer pointed out, analyses of these small numbers by RNA-seq may easily lead to unsuitable conclusions with less reproducibility. Therefore, we drew a roadmap to obtain a firm conclusion. We first examined whether there are transcriptomic differences between different clonotypes of hTERT-specific T cells. Using the results of scRNA-seq, we concluded that there appeared to be no obvious differences and then we decided to treat the different clonotypes as combined hTERT-specific T cells. To observe differences between the clonotypes, we perhaps had needed more Ag-specific T cells in the dataset. Consequently, the goal of the analyses in this study was to identify

transcriptomic signatures of total hTERT-specific T cells. However, the numbers of hTERT-specific T cells detected in the RNA-seq analyses remained too low to calculate differentially expressing genes in these cells. Because we noticed that hTERT-specific T cells were a certain subpopulation, we decided to analyze differentially expressing genes of the subpopulation compared with others. The subpopulations were determined by k-means clustering following t-SNE dimension reduction (Fig. 6e–h). We believe that this approach diluted inadvertent factors that might occur because of the low number of hTERT-specific T cells and produce more reliable results. Because we did not conduct scRNA-seq of other hTERT patients, it remains unclear whether our result is shared among the different patients. Instead, we have provided more information about the AFP-specific T cells from D3 in the A2 transcriptome data. We showed that AFP-specific T cells from D3 were plotted near the hTERT-specific T cells of A2, which suggested a transcriptomic vicinity to hTERT T cells. This result supported that our conclusion could be generalized.

To address the reviewer's concern about whether we could conclude that transcriptomic features of the same clonotypes changed over time, we carefully reviewed the transcriptomic data and Fig. 6f. Although we observed accumulation of hTERT-specific T cells in the combined t-SNE plot, there appeared to be a different distribution between 1 and 5 years as shown below. K-means clustering also roughly distinguished 1-year from 5-year (clusters 5 and 4), which may have resulted in the different transcriptomic profiles (Fig. 6g, h). However, emphasizing this difference and

discussing transcriptomic changes is a possible overstatement here because of the small cell number and sample size as noted by the reviewer.

We have changed the possible misleading sentences as shown below:

Lines 47–49

“These T cells belonged to a CD8⁺ T cell population that expressed a typical memory subset with high expression of IL7R, SELL, and NOSIP along with promotion of the AP-1 transcription factor network after 5 years.”

Lines 181–182

“To further understand the characteristics of the peptide-specific T cells, we tracked the specific TCR chronotypes along with their transcriptome data over years after vaccination using the single cell RNAseq technique.”

7. Given that the conclusion is that vaccines can generated such long term memory responses, it would be of interest to increase the number of patients studied and track the transcriptomic profile of CMV specific cells at the same time points to see if these changes are unique to vaccine specific TCRs, but not CMV-specific lymphocytes.

Response: The idea suggested by this comment to compare vaccine-specific T cells with CMV-specific T cells is an elegant approach to further characterize the vaccine-specific T cells. Because remaining frozen peripheral blood samples of A2 were stored and available, we conducted an experiment to address this comment. We first detected CMVpp65-specific CD8 T cells among A2 PBMCs using an HLA-A24:CMVpp65₃₂₈₋₃₃₆-PE tetramer (MBL Co., Japan) and sorted them out as single cells. Using hTEC10 technology (Kobayashi, Mizukoshi et al. Nat Med 2013), we analyzed TCR α/β chain sequences. Paired sequences were successfully obtained in six cells out of the 11 sorted single cells. The results homogeneously indicated [TRAV24/TRAJ23/CDR3: CAPTTGGKLIF] and [TRBV9/TRBJ1-1/TRBD1/CDR3: ASSVGQGAYTEAF] in all cells. We assumed this TCR α/β as specific for CMV in A2 and then identified CMV-specific T cells in the combined CD8 t-SNE plot (Supplementary Fig. 7). We only found three CMV-specific T cells (1 at 1 year and two at 5 years). Therefore, we could not discuss how the gene expression of CMV-specific T cells as compared with that of the vaccine-specific T cells. Instead, we noted it was obvious that the CMV-specific T cells were distinguished transcriptomically from the vaccine-specific T cells because these cells were located in the “regular” CD8 region.

It is of interest to pursue transcriptome features in the other patients. Unfortunately, the patient samples are almost exhausted because our previous experiments, patient death, or discontinuation of attending our hospital. We are planning to conduct a phase 2/3 study of the peptide vaccine and carry out the same scRNA-seq experiments with an increased patient number to address this issue.

We have added the text below to the manuscript:

Lines 225–229

“To compare the overall gene expression features of the peptide-specific T cells, we combined the scRNA-seq data of 1-year CD8⁺ T cells and 5-year CD8⁺ T cells that included peptide-specific T cells. We found accumulation of peptide-specific T cells in the t-SNE plot, although the clusters that contained these cells had rarely overlapped, which suggests that these cells shared transcriptomic features. However, CMV-specific T cells were located in the rest of the plot (Fig. 6e–f, Supplementary Fig. 7).”

8. In Figure 6d and e, it would be preferable to provide clustering for both 1 year and 5 year all together instead of providing separately so clusters and TCRs detected in specific clusters can be directly compared.

Response: To address this comment, we carefully considered how to integrate 1- and 5-year data together; and how we could introduce our findings from each dataset into the combined plots. Because we believed that it was more interesting to also observe the existing clusters generated in each dataset in the combined plot with the TCR information rather than to generate a new set of clusters in the combined plot (this is shown in response to Reviewer #2’s major comment #6), we combined 1- and 5-year CD8 T cells together and carried out the t-SNE with the existing clusters that remained to determine their transcriptomic locations. The result has been added to Fig. 6e. We obtained partially blended and independent accumulation of the clusters. Briefly,

CL1_1-year was roughly overlapped with CL1_5-year and CL3_1-year was overlapped by CL2_5-year. CL2_1-year appeared to be independent, but partially overlapped with CL3_5-year. CL2_1-year and CL2_5-year, in which the most peptide-specific TCRs were detected, appeared to be independent from each other. However, TCR plotting showed accumulation of these TCRs across the two clusters (Fig. 6f) as discussed above. We have added the sentence below:

Lines 227–228

“We found accumulation of peptide-specific T cells in the t-SNE plot, although the clusters that contained these cells had rarely overlapped, which suggests that these cells shared transcriptomic features.”

Minor points:

1. Figure legend 1d is missing

Response: We appreciate this comment. We provided the explanation for 1d but not “(d).”

We have added “(d)” to the figure legend.

2. Figure 3g FMO control for Tet⁺ and Tet⁻ could be included in the Supplementary data to ensure that the expression of PD-1 reported has been well compensated.

Response: We did not include FMO controls in the original experiment. We appreciate the reviewer’s concern about potential compensation problems, especially within rare tetramer⁺ fractions. The gating strategy was as follows. First, we drew a positive gate

on the tetramer⁻ fractions, which showed a clear distinction, and then used the same cutoff intensity in tetramer⁺ populations. As the reviewer indicated, this might lead to less “accurate” results in terms of the exact frequencies of PD-1/CTLA-4 positivity within the tetramer⁺ fractions. However, we assumed that we would have “precise” results of the relative frequencies using the same gates under the appropriate compensation setting (described later). To further address this comment and validate our results, we repeated the same analysis using the remaining refrozen A7 PBMCs. We replicated this experiment using the same protocol with identical antibodies, which included FMOs this time, and the conserved cytometer settings with updated laser delays and compensation settings over time. The experiment is shown in Supplementary Fig. 2, where we found no positive cells in the FMO controls and no compensation issues with the same cutoff strategy. The frequencies are also faithful to our original analysis. These results reinforced our data in Fig. 3.

We have added the reference to Supplementary Fig. 2:

Lines 121–123

*“In addition to ELISpot assays, we also evaluated peptide-specific T cell induction by flow cytometry using hTERT₄₆₁, AFP₃₅₇, AFP₄₀₃, and MRP3₇₅₉ tetramers along with other phenotypic markers (see **Supplementary Fig. 2**).”*

3. Lines 215-217, please clarify which peptides specific cells are being compared.

Response: We combined hTERT- and AFP-specific T cells to identify differentially expressed genes in peptide-specific T cells. We have added the explanation as shown below.

Lines 234–236

“We next identified significantly differentially expressed genes in combined peptide-specific T cells, which included both hTERT461 and AFP357 detected in 1- or 5-year samples, compared with other CD8 T cells of 1- and 5-year samples.”

4. Line 258: add “in selected patients”

Response: We have added the phrase below:

Line 261–262

“The vaccines maintained sustainable immune responses beyond 10 years without continuous administration of the peptides in selected patients.”

5. Lines 275 and 276 differences between finding two cells and one cell at different time points does not support that this TCR was involved in controlling disease.

Response: As the reviewer pointed out, this was apparently an overstatement. Therefore, we deleted the sentence.

Line 275

~~“which convinced us that the highly functional D3.14 TCR played an important role in controlling the tumor in the D3 patient.”~~

6. Technical details flow cytometry, including compensation and how gating was done strategy and negative controls (specific use of FMO controls, if they were used).

Response: In this study, we analyzed cells on a BD FACSAria II in a consistent manner with compensation settings. We perform compensation using single fluorochrome-stained cells or beads. First, we adjust PMT voltages to avoid excessively weak or saturated fluorescence signals for each color. Here, we used cell samples even when using compensation beads in the following procedure. Because the BD FACSDiva provides easy automated compensation, we used this function for a rough adjustment. Additionally, we fine-tuned the compensation manually. Briefly, when we compensate fluorochrome A emission to fluorochrome B detection, we adjust the percentage compensation so that MFIs of the fluorochrome B channel in fluorochrome A+ and fluorochrome A– fractions are equal with the fluorochrome A single-stained sample. We repeat this process for every combination of fluorochromes until we achieved optimal compensation.

The gating strategies in this experiment are shown in Supplementary Figs. 2 and 3. Although we did not usually use FMO controls at the beginning, we believe these were still reliable to evaluate the relative expression of the molecules in this study. Our verification experiment also supports the results obtained by FCM in this study as described above (Supplementary Fig. 2).

7. Did the authors analyze the presence of responses in CTL- patients as well as negative control?

Response: We appreciate this comment because a lack of negative controls might reduce the impact of the results of immune responses at each time point. In the ex vivo analyses, we used cognate peptides and a HIVenv-derived peptide as a negative control. The negative control wells showed no spots or less than 10. We have added these data in Fig. 3. For the preculture assay in which we cultured patient PBMCs in the presence of the corresponding peptide for 7 days before the ELISpot assay, we did not include negative control wells because there was not enough PBMC samples.

We also conducted an ELISpot assay to assess CTL(-) patients from hTERT and AFP trials. because there were no CTL (-) patients who had survived for 5 years, we only examined PBMCs collected at 1 year after vaccination. In the AFP trial, D13 alone was in a good condition to collect blood samples. D13 lymphocytes showed minimal responses to AFP peptides even after preculture. In the hTERT trial cohort, the five surviving individuals were assessed to determine whether hTERT-specific immune responses occurred. Three of them (A7, A10, and A12) did not have hTERT-specific immune responses, whereas A9 and A13 exhibited hTERT-specific immune responses. These unexpected immune responses were explainable by examining the initial immune induction profiles, although this would be provisional and exploratory. A9 was a borderline CTL(-) case where 2.5 immunospots before the vaccine had increased up to 11 spots after immunization, which did not meet the criteria for CTL induction. The

increased immune responses were maintained at 1 year as judged by the ex vivo ELISpot result of 12 immunospots and might be expanded by preculture. A13 had already shown several immunospots before the vaccine. Nevertheless, the immune response was not boosted by treatment (Pre: 12.5 → Post: 14.5). The baseline response had remained and expanded under the in vitro condition. The other examples of high baseline responses were A3 (Pre: 17 spots) and A14 (Pre: 12.5 spots), which were enhanced by the vaccination (both had 35 spots) and classified as CTL induction (+). These patients had durable survival periods (6.1 y and 7.8 y, respectively) compared with A13 whose survival was approximately half (3.0 y). This finding also supports our definition of CTL(+) as useful to predict prognosis because it distinguished A3 and A14 from A13. ELISpot results of CTL(-) patients 1 year after vaccination and the pre- and post-ELISpots of A9 are shown below:

ELISpot results of CTL(-) patients from hTERT and AFP trials Patient PBMCs were incubated in the presence of the peptide on ELISpot plates. In preculture(+) experiments, PBMCs were cultured for 7 days in the presence of the Ag peptide before the assay. IFN γ was captured and visualized. Each spot was considered as an Ag-specific T cell. Numbers of spots are shown in a bar graph. Each experiment was conducted in duplicate. Error bars: s. e. m. **(Left)** Immunospots are shown, which were obtained in the experiment of judgement of CTL(+) or CTL(-). A9 was judged as CTL(-), but there were some increases of immunospots after immunization **(Right)**.

Reviewer #3 (Remarks to the Author):

The statistical methods applied by Mizukoshi et al. in this manuscript are appropriate and support the conclusions of the study. Sample sizes are admittedly modest, however considering the nature of the study I support that the analysis of the available data is suitable and the biological significance of the findings should be evaluated despite modest sample sizes.

We have placed numbers in front of each comment.

1. The authors have performed survival analysis using log ranked test, reporting p-values and hazard ratios. I would recommend the authors to perform multivariate survival analysis taking into account the traditional predictors of outcome and incorporate the results in the manuscript.

Response: As the reviewer indicated, multivariate analyses are of great interest to exclude any confounding factors that might affect the understanding of the results of our study. Therefore, we performed multivariate survival analyses with some traditional prediction markers as parameters. We adopted age, clinical stage (The liver cancer study group of Japan), hepatic reserve (Child–Pugh score), and serum AFP level as well as CTL induction. We analyzed the hTERT and AFP vaccine cohorts using the Cox regression hazard model in R software with the package “survival” (<https://github.com/therneau/survival>). We confirmed positive CTL induction and an

early clinical stage correlated with prolonged overall survival periods in both cohorts as shown in Supplementary Table 5. We observed some extreme hazard ratios due to the small sample numbers. Therefore, we used them as provisional values. We plan to conduct phase 2/3 trials to provide further evidence of the survival benefits of the peptide vaccines, which will also allow us to confirm the prognostic factors. We have added the sentences below to the manuscript:

Lines 115–117

“CTL(+) and an early clinical stage²¹ appeared to be significant factors for survival in the multivariate analysis (Supplementary Table 5). However,”

2. The authors should mention the test used for the differential gene expression analysis in the figure legends (Figure 6 and the associated supplemental figures) and in the Methods. Also, in the same analyses the p-values don't seem to have been adjusted for multiple testing and may not withstand multiple testing correction but I support that this is still valuable information of the differences in the comparison groups and suitable for publication.

Response: These are very helpful comments and allow us to explain more about the analyses that we adopted in the scRNA-seq experiments. We used the Cell Ranger pipeline from 10x Genomics, which already includes multiple comparison components. For the differential gene expression analysis to test mean expression of genes between clusters, this software performs the exact negative binomial test: sSeq method (Yu,

Huber & Vitek 2013) or the fast asymptotic negative binomial test used in edgeR (Robinson & Smyth 2007). The tests generate p-values adjusted for multiple testing using the Benjamini–Hochberg procedure to control FDR. The detailed algorithms have been released to the public as shown below:

<https://github.com/10XGenomics/cellranger/blob/master/lib/python/cellranger/analysis/diffexp.py>

We have added an explanation in the methods as shown below:

Lines 446–450

“For the differential gene expression analysis testing mean expression of genes between clusters, the exact negative binomial test or the fast asymptotic negative binomial test was performed generating p-values adjusted for multiple testing using the Benjamini-Hochberg procedure to control FDR (<https://github.com/10XGenomics/cellranger/blob/master/lib/python/cellranger/analysis/diffexp.py>).”

3. In supplementary figure 5 and 6 the range of the p-values should be indicated as graphical key or in the legend text. The authors have appropriately provided details of the other tests used in the relevant figure legends and there is a statement in the Methods mentioning the software used to generate the statistics.

Response: We have added graphical annotations that indicate p-values and improved the visibility of the pathway analyses in Supplementary Figs. 5 and 6.

REVIEWERS' COMMENTS

Reviewer #1 (Remarks to the Author):

The manuscript has been improved.

Reviewer #2 (Remarks to the Author):

The authors have performed a considerable amount of experimental work and have addressed all but one of the concerns raised. No additional patients were included in the scRNAseq studies due to sample limitations. However, one of the major conclusions of the article is based on this scRNAseq experiment which only includes data from two patients. This should be pointed out in the abstract and elsewhere to make the reader aware of this fact.

Reviewer #3 (Remarks to the Author):

The authors have addressed my concerns in their revised manuscript.

Responses to REVIEWERS' COMMENTS

We answered the reviewers' comments in a point-by-point basis.

Reviewer #1 (Remarks to the Author):

The manuscript has been improved.

Reviewer #2 (Remarks to the Author):

The authors have performed a considerable amount of experimental work and have addressed all but one of the concerns raised. No additional patients were included in the scRNAseq studies due to sample limitations. However, one of the major conclusions of the article is based on this scRNAseq experiment which only includes data from two patients. This should be pointed out in the abstract and elsewhere to make the reader aware of this fact.

As the reviewer indicated, we put descriptions of the limitation of our study in terms of sample number as below;

Line 50-51

"in two patients"

Line 306-307

"We note that our scRNA-seq results were based on the limited samples that only included two patients due to sample availability."

Reviewer #3 (Remarks to the Author):

The authors have addressed my concerns in their revised manuscript.